# Biodegradable Starch/Chitosan Foam via Microwave Assisted Preparation: Morphology and Performance Properties

**DOI:** 10.3390/polym12112612

**Published:** 2020-11-06

**Authors:** Xian Zhang, Zhuangzhuang Teng, Runzhou Huang, Jeffrey M. Catchmark

**Affiliations:** 1Co-Innovation Center of Efficient Processing and Utilization of Forest Products, College of Materials Science and Engineering, Nanjing Forestry University, Nanjing 210037, China; shinezhang17@163.com (X.Z.); RowanZJ@163.com (Z.T.); 2Department of Agricultural and Biological Engineering, The Pennsylvania State University, University Park, State College, PA 16802, USA

**Keywords:** chitosan, potato starch, microwave, foam, orthogonal experiments

## Abstract

The effects of chitosan (CTS) as the reinforcing phase on the properties of potato starch (PS)-based foams were studied in this work. The formic acid solutions of CTS and PS were uniformly mixed in a particular ratio by blending and then placed in a mold made of polytetrafluoroethylene for microwave treatment to form starch foam. The results showed that the molecular weight and concentration of CTS could effectively improve the density and compressive properties of starch-based foams. Furthermore, orthogonal experiments were designed, and the results showed that when the molecular weight of CTS in foams is 4.4 × 10^5^, the mass fraction is 4 wt%, and the mass ratio of CTS–PS is 3/4.2; the compressive strength of foams is the highest at approximately 1.077 mPa. Furthermore, Fourier transform infrared spectroscopy analysis demonstrated the interaction between starch and CTS, which confirmed that the compatibility between CTS and PS is excellent.

## 1. Introduction

Global pollution is increasingly becoming a serious concern, causing extreme climate changes and natural disasters. The huge pollution of traditional plastic products in the environment has extensively raised people’s concern. Traditional plastics such as polystyrene (PS), soft (hard) polyurethane, polyvinyl chloride, and polyolefin are widely used as the matrix of foaming materials, but due to the poor operability and degradability of these plastics, most of the foaming materials are wasted, leading to increasingly serious white pollution. With people’s increased awareness of environmental protection, considerable attention has been paid to the development of environmentally friendly materials. Therefore, bio-based and biodegradable polymers as replacement materials for traditional nonbiodegradable plastics have been extensively studied [1,2,3,4]. Biodegradable plastics in the natural environment are decomposed into carbon dioxide (CO_2_) and oxygen (O_2_) by the action of water (H_2_O), light, and microorganisms, and they hardly pollute the environment. Conventional plastics are nondegradable, which degrade slowly and require a long time, even hundreds of years, to degrade completely. They also impact the soil negatively during this time. Therefore, the exploration of degradable plastics that can replace traditional plastics holds great practical significance.

Starch, a rich polysaccharide substance found on earth, is an important raw material for the synthesis of biodegradable plastics. Starch has excellent biodegradability, and it can be decomposed by microorganisms in the natural environment and finally be metabolized into H_2_O and CO_2_. Starch-based plastics are beneficial in being nontoxic, harmless, widely available, and cheap and also possess relatively superior properties. Use of these plastics effectively can alleviate the problem of white pollution and the crisis of biochemical energy shortage [5,6]. Biodegradable porous materials prepared from starch are being widely used in the fields of food packaging and medical tissue engineering [7]. To ensure complete degradability of the system, various biodegradable natural polymers such as cellulose, lignin, and chitosan (CTS) and polyesters such as polylactic acid (PLA), polycaprolactone, polybutylene succinate, and polyvinyl alcohol have been blended with starch in an attempt to synthesize biodegradable plastics [8,9,10,11,12,13]. Loercks et al. [14] successfully developed a degradable thermoplastic starch-based product by compounding starch with a hydrophobic biodegradable polymer, which greatly enhanced its aerobic degradation rate. Ferri et al. [15] reported that the addition of glycerin, polyethylene glycol, and other plasticizers to the blend of thermoplastic starch (TPS) and PLA could significantly reduce the cost, enhance mechanical performance, and broaden the application ranges of PLA-TPS blends. Bénézet et al. [16] prepared fiber-reinforced foams by the extrusion method and reported that the addition of fibers increases the expansion index and significantly reduces the water absorption of starch-based foams; the foams with 10% hemp fibers presented excellent mechanical properties.

CTS consists of β-(1,4)-linked 2-amino-deoxy-d-glucopyranose, and it can be synthesized by deacetylation of chitin [17,18]. It is soluble in acidic solutions because of the protonation of its –NH_2_ group at the C–2 position of the glucosamine unit. Owing to its biodegradability and unique physicochemical properties, it is widely used in the preparation of hydrogels, films, fibers, or sponges and in the field of biomedicine [19]. Although CTS is a novel substance that possesses excellent film-forming potential, permeability, and antibacterial properties, it has many deficiencies. Dang et al. [20] prepared TPS or CTS films by using different CTS concentrations and confirmed that CTS is distributed on the surface of films. Due to the high crystallinity and hydrophobicity of CTS and its ability to interact with starch-forming intermolecular hydrogen bonds, CTS on the membrane surface improves the water vapor and oxygen barrier properties of materials and decreases the hydrophilicity of the membrane surface.

Starch and chitosan are natural high molecular polymers with excellent biocompatibility and biodegradability, which can be completely degraded without causing harm to the environment [21]. Currently, most research has focused on the synthesis of the starch–CTS membrane [22,23,24,25], and the application of CTS-starch foams in the field has been rare. The foam prepared by starch and chitosan is a kind of biological material, which is easy to be processed and molded. The chitosan in foam can promote the formation of blood coagulation and thrombosis, inhibit the growth of various bacteria and fungi, and promote the repair of damaged tissues [26]. Therefore, the foam is expected to be a good hemostatic material and packaging material for food and medicine. In this paper, we aimed to manufacture a new foam material by using CTS and potato starch (PS) in a polytetrafluoroethylene mold for microwave treatment, analyze the effects of molecular weight and concentration of CTS on the morphological characteristics and functional properties, such as physicochemical, mechanical, and thermal properties, of starch-based foam, and determine the optimum proportion of CTS and PS in foams to achieve excellent performance. We hope to enable further development of starch-based foams for use as active materials in biomedical applications as well as food and drug packaging.

## 2. Materials and Methods

### 2.1. Materials

PS (with approximately 75% amylopectin and 25% amylose) and CTS (low viscosity ≤200 mPa·s; medium viscosity: 200–400 mPa·s; high viscosity ≥400 mPa·s) were purchased from Shanghai Meryer Company (Shanghai, China). Formic acid (96%) was purchased from Shanghai Macklin Company (Shanghai, China).

### 2.2. Sample Preparation

(1) Determination of the molar mass of CTS

Acid hydrolysis was used to determine the molecular weight of three CTS samples used in this study. The dried CTS powder was dissolved in 0.2 M acetic acid (HAc)/0.1 M sodium acetate (NaAc) solvent (pH = 4) and different concentrations of the solution were prepared; CTS powder was dissolved completely in the solvent by moderately heating the solution at 40 °C. The 10 mL solvent and cooled CTS solution were injected into the Ubbelohde viscometer (capillary diameter = 0.8 mm) (Shanghai Liangjing Glass Instrument Factory, Shanghai, China), and the vertical state Ubbelohde viscometer was placed in a constant temperature water bath at 30 °C for 15 min. The outflow time of solvent and CTS solution (*t*, *t*_0_) was measured with a stopwatch, and the mean value of the three measurements was recorded (the difference was not more than 0.2 s).

(2) Preparation of CTS-starch based foams

The starch-based foaming materials were formulated using CTS and PS by heating in a microwave oven. A WD900ASL23-2 Galanz microwave oven (Galanz Shanghai Company Limited, Shanghai, China) with microwave emission frequency of 2450 MHz was used to heat CTS-starch compounds under microwaves at 100% power of 900 watts. The container carrying the CTS-starch mixtures was located at the center of the microwave chamber and irradiated by microwaves for the same time (40 s). Different CTS-starch mixtures were packed in special containers and the specific experimental process is shown in Figure 1. After microwave treatment, the samples were conditioned overnight at 25 °C before further characterization.

(3) Experiment design

The comprehensive experiment was designed with three factors at three levels respectively (3^3^ = 27 combinations, without considering the reproducibility of experiments), namely the viscosity of CTS (A), the mass fraction of CTS (B), and the weight ratio of CTS solution/starch (C). Table 1 shows the experimental design under different factors and levels. 

On the basis of the comprehensive experiment, orthogonal test analysis and multi-factor analysis of variance were carried out to explore the interaction factors’ influence on the mechanical properties, which would further help the experiment to determine the best process parameter of each level and to determine the relative importance of individual parameters and the combination of process parameters with high performance.

### 2.3. Characterization

(1) Calculation of CTS molecular weight

When the viscosity of polymer in a wide range of molecular weight is measured with the same viscometer, the relative value of solution viscosity to solvent viscosity or the relative viscosity, *η_r_*, is expressed as *t/t*_0_ (*t*_0_ and *t* are the outflow time of dilute solution and pure solvent, respectively, under the same temperature and measured using the same viscometer,). The specific viscosity, *η_sp_*, which reflects the internal friction effect between polymer and polymer and between pure solvent and polymer, is expressed as follows:*η_sp_ = (η − η*_0_*)/η*_0_* = (η*_0_*/η*_0_*) − *1* = η_r_ − *1**(1)

The intrinsic viscosity [*η*] is defined as the reduced specific viscosity (*η_sp_/c*) or the relative viscosity of the inherent relative viscosity (*ln η_r_/c*) of a solution in the case of infinite dilution of solution mass concentration (*c*), which is expressed as follows: 
(2)
$$\left[\eta \right]=\underset{c=0}{\lim}\eta_{sp}/c=\underset{c=0}{\lim}\ln\eta_{r}/c$$


The intrinsic viscosity [*η*] of polymer solution exhibits a specific relation with the molecular weight of polymer at a specific temperature. According to the Mark–Houwink–Sakurada (MHS) equation, the following empirical formula is used for calculating [*η*] and the molecular weight of polymer solution [27,28,29]:
(3)
$$\left[\eta \right]=KM^{a},$$

where *M* is the molecular weight, *K* and *α* are constants for given solute–solvent system and temperature, respectively. In this study, *K* and *α* were 6.59 × 10^−3^ and 0.88, respectively.

The relationship between *ln η_r_*/*c* (or *η_sp_/c*) and mass concentration *c* is linear. Thus, a straight line can be drawn by plotting *ln η_r_*/*c* (or *η_sp_/c*). When *c* is extrapolated from Equation (2), the line should cross a point on the y-axis; the intercept is the intrinsic viscosity [η]; the molecular mass of polymer can be calculated using Equation (3).

(2) Morphology of starch-based foams

The starch-based composite structure was investigated using scanning electron microscopy (SEM) (FEI 200 Quanta FEG, pressure 0.9 Torr) (FEI Company, Hillsboro, OR, USA). Foam samples were sputter-coated with platinum for 20 min after cutting with a razorblade into 3-mm thick slices, and the transversal cross-section area was examined using SEM.

(3) Pore size, porosity, and density of CTS–PS foams

Image analysis (Image-J) software (National Institutes of Health, Bethesda, MD, USA) was used to measure the average diameters of nearly 100 pores. Plot of pore size distribution enabled the comparison of cellular structure for different formulations of starch foams. The finished foam was cut into five samples of 24 × 24 × 32 mm^3^ size, and then the exact size of the sample was measured to ensure that the error was less than 0.02 mm. The sample mass was weighed with an electronic balance to make the result accurate to 0.001 g, and finally the density of the material was calculated using the following formula:
(4)
$$\rho =\frac{M}{V}\times 10^{3}$$

where *ρ* = density (g/cm^3^), *M* = foam weight (g), *V* = foam volume (cm^3^)

(4) Mechanical property

According to the standard GB/T 8813-2008, the compressive strength of foam materials was tested using a CMT 4000 universal testing machine (Mechanical Testing & Simulation Company, Eden Prairie, MN, USA). The vertical direction of foam growth was consistent with the direction of compression. The compressive strength of the foam material is the ratio of the maximum compressive stress to the original cross-sectional area of the sample, when the relative deformation of the material is less than 10%, which is expressed in kPa.

The specific formula is as follows:
(5)
$$P=F/S=F\times 10^{4}/\left(l\times w\right)$$

where *P* = compressive strength (kPa), *F* = compressive stress (kN), *S* = cross-sectional area (cm^3^), 
l
 = length (mm), 
w
 = width (mm).

(5) Fourier transform infrared analysis

The dried samples and potassium bromide were ground into a fine powder in an agate mortar and pressed into tablets. The Fourier transform infrared (FTIR) spectrum of material was measured using the VERTEX 80 V type FTIR spectrometer (German Bruker company, Karlsruhe, Germany) and the spectrum at a range of 4000–400 cm^−1^ was acquired for every sample. The processing software OMNIC 8.0 (Thermo Nicolet Corporation, Fitchburg, WI, USA) was used to perform automatic baseline correction of the collected infrared spectrum to ensure its baseline level.

(6) Thermal characterization

The total mass loss on a TG209F3 analyzer (Netzsch Corporation, Bavarian, Germany) was assessed using the thermogravimetric (TG) method. For the 6 mg samples dried for 24 h at 70 °C, the experimental temperature was set as 30 ± 3 °C to 800 °C in nitrogen atmosphere. The heating rate was 10 °C/min, and the experimental flow rate was 30 mL/min.

(7) Differential scanning calorimetry (DSC) test

The glass transition temperature of sample was evaluated by a 200 F3 Differential Scanning Calorimeter (Netzsch Corporation, Bavarian, Germany). The dried sample of around 3 mg was placed in an aluminum crucible and sealed. An empty aluminum box was used as a control and nitrogen was used as purge gas. The temperature ranged from 20 to 180 °C and the heating rate was 10 °C/min.

(8) Solubility of foams in water

The samples were dried in a drying oven at 60 °C for 24 h and then soaked in deionized water at room temperature. The shape of the sample in water was recorded until it dissolved.

## 3. Results and Discussion

### 3.1. Molecular Weight of CTS

The viscosity of CTS solution at a particular concentration directly reflects its molecular weight. Assuming other factors to be fixed, the higher the relative molecular weight of CTS, the higher will be the viscosity of the CTS solution, and the smaller the relative molecular weight is, the smaller will be the viscosity of the CTS solution. Figure 2 illustrates the trend line of the CTS sample, with a low viscosity extrapolated to the intersection point of the straight line and y-axis to be 502.39, implying that the characteristic viscosity of the CTS, [*η*] was 502.39, and the molecular weight of the CTS sample calculated according to the MHS equation was 3.5 × 10^5^. Likewise, the molecular weights of other two types of CTS were also estimated through the same calculation; Table 2 summarizes the values of the molecular weight. The correlation coefficient (R^2^) of CTS with different viscosities was greater than 0.95, which indicates the reliability of the calculated results. The molecular weight of CTS with viscosity between 200 and 400 mPa·s was 4.4 × 10^5^ and that of CTS with a viscosity of 400 mPa·s was 5.2 × 10^5^ (Table 2), and the difference was significant; the trend was completely consistent with the above-mentioned.

### 3.2. Morphology of Starch-Based Foams

Figure 3 presents SEM observations of the starch-based foam prepared using the molecular weight of CTS as a variable. Figure 3a,b indicate that the size and distribution of pores on the foam surface produced by microwave radiation are not as apparent as those observed in the inner layer structure. The pores on the surface layer were small and unevenly distributed, whereas the bubbles in the inner structure were significantly larger and more uniform. Starch-based foams made of CTS with different molecular weights also demonstrated distinct characteristics. Comparison of the three types of foam surface structure (Figure 3a,c) revealed the largest number of pores in the foam with CTS of molecular weight 4.4 × 10^5^ (CTS-4.4/PS) (Figure 3e); the quantity of pores in foam with the CTS of molecular weight 3.5 × 10^5^ (CTS-3.5/PS) was relatively smaller (Figure 3a) than that in the CTS of molecular weight 5.2 × 10^5^ (Figure 3c). In short, the increasing order of the number of pores according to the SEM observation is as follows: Figure 3e > Figure 3a > Figure 3c. The trend observed in the pore quantity on the inner structure was exactly the opposite of that observed in the surface morphology. From the SEM images in Figure 3b,d,f, we can clearly observe that the starch-based foam made from medium-viscosity CTS (Figure 3d) has the largest number of pores, relatively smaller pore size (a rough comparison by the yellow circles in Figure 3b,d,f), and relatively regular bubble morphology compared with other foam types.

### 3.3. Pore Size, Porosity, and Density of CTS–PS Foams

The pore size and porosity were measured using Image-J software, and Table 3 presents the specific values. We observed that at a fixed amount of CTS solution and with an increase in the starch mass, the bubble porosity and porosity decrease in all the CTS–PS composites. The decrease in foam hole and porosity in the same volume mold with increase in starch content is quite reasonable. However, the pore size and porosity of foams with different levels of starch quality also changed with changes in the molecular weight of CTS. When the mass ratio of CTS solution and starch was 3:4, the bubble size and porosity of CTS-4.4/PS foam composites were the smallest (281.7 μm and 55.39%, respectively), whereas the difference in bubble size between CTS-3.5/PS and CTS-5.2/PS foams was not significant, which indicates that the molecular weight of CTS affects the starch-based foams to a certain extent; however, the effect is limited.

According to the density of the starch-based foam shown in Figure 2, when the mass fraction of CTS solution was 4%, the density of foams prepared from CTS solution with the same molecular weight gradually increased with the increase in starch content. However, when the mass ratio of CTS solution and starch was constant, the change was different from that observed earlier, and the density of CTS-4.4/PS foams was apparently higher (Figure 4). Figure 4 shows the influence of mass fraction of different CTS solutions on starch-based foams, when the mass ratio of CTS/PS was 3/4.2. Notably, the mass fraction of CTS solution had only a minor effect on the density of the entire foam material, whereas the molecular weight of CTS exerted a greater effect on the same. Table 4 summarizes the specific values of all composites. Combined with the aforementioned SEM observations on the pore size and porosity of foam materials, we can assume that by using a particular mass fraction of the CTS solution, the pore structure and density of materials can be improved by appropriately increasing the starch content. This is because when the starch content increases, the formic acid in the CTS solution and PS generates more esterification reactions in the microwave, and since the volume of the mold is constant, it appears as a prepared foam in a macroscopic view. Thus, the pore size and porosity gradually decrease, and correspondingly, the density of foams gradually increases.

### 3.4. The Compressive Property of Foams

Biodegradable foam must possess definite mechanical strength to maintain its integrity during transportation, and the density and compressive strength of foams are somewhat correlated. Figure 5 illustrates the curves for the compressive strength of PS-based foams with change in the molecular weight of CTS and mass ratio of CTS/PS to 3/4.2 at 10% compressive ratio. The variation trend of these curves was basically consistent, as shown in the figure. The CTS-4.4/PS foams exhibited greater density and superior compressive performance compared with CTS-3.5PS and CTS-5.2PS foams. The increase in the density of the foam increases the bearing capacity of a single bubble hole and the effective bearing area within the unit section, which is generally reflected in the increased overall compressive strength of the foams.

The foams composed of CTS and starch exhibit such strength mainly because of the formation of a high intermolecular hydrogen bond between NH_3_^+^ in the main chain of CTS and OH^+^ in the starch molecule. The amino group of CTS is easily protonated to form NH_3_^+^ in formic acid solution, and the ordered crystal structure in the starch molecule is destroyed during the microwave irradiation treatment, which leads to the formation of a hydrogen bond between the hydroxyl group of starch and amino group of CTS. In addition, electrostatic interactions between NH_3_^+^ in CTS and phosphate groups in PS may occur. Phosphorus, the most important element in PS, exists in the form of a covalent bond in PS. Glucose-6-phosphate (pKa_1_ = 0.94, pKa_2_ = 6.11) is the main structure of esterified phosphate in PS, and its acidity is stronger than that of orthophosphate [30,31]. Therefore, phosphate groups in PS possess a negative charge in aqueous solutions and do not combine with other negatively charged substances. The amino groups in the CTS solution possess a positive charge after being protonated and interact with the negatively charged phosphate groups in PS, which further improves the compressive performance of the entire CTS-PS foams. Under the condition of 4.2 wt% CTS, the compressive strength of CTS-PS foam increased with the increase in starch quality and the maximum compressive value was obtained at a mass ratio of 3/4.2. The molecular weight of CTS demonstrated a certain influence on the compressive strength of the starch-based foams. Among the foam composites prepared from CTS with three different molecular weights, the CTS-4.4/PS foams had the highest compressive strength value and those of CTS-3.5/PS and CTS-5.2/PS foams were extremely close (Table 5). The interaction between starch and CTS has been reported to enhance the functional properties [32]; however, when the number of NH_3_^+^ species increases beyond a critical value, synthesis of uniform CTS/PS foams becomes challenging and results in weak interaction at the boundary and poor compressive performance.

To determine the effect of each factor on the foam compression performance, we further designed an orthogonal experiment and variance analysis. Table 6 presents the experiment results. According to range (R) analysis, R (C) > R (A) > R (B), and combined with the variance analysis in Table 7, we found that factor A (CTS molecular weight) and factor C (the mass ratio of CTS to starch) were all significantly affected, whereas factor B (CTS mass fraction) displayed no significant difference in the experimental results (P > 0.05), which indicates that the mass fraction of CTS fluctuating slightly above 4 wt% does not considerably affect the compressive performance of the foams. According to the results of intuitive and variance analyses, the optimal combination of foam preparation was A_2_B_2_C_3_, implying that the CTS molecular weight of 4.4 × 10^5^, CTS mass fraction of 4%, and the CTS/PS mass ratio of 3/4.2 are optimal for foam synthesis. We verified the compression performance of the CTS starch-based foam under these conditions, which was higher (1.077 mPa) than those observed in other conditions.

### 3.5. FTIR Analysis

To study the interaction between CTS and PS, an infrared spectrum of each material was obtained. Figure 6 illustrates an extremely wide stretching vibration band at 3413 cm^−1^ for the dried PS corresponding to the characteristic absorption peak of the hydroxyl group in the starch. A moderate intensity peak at 2930 cm^−1^ was associated with –CH_2_ antisymmetric stretching vibration, and an absorption band at 1647 cm^−1^ was attributed to water absorption in the amorphous region of PS; the absorption peak near 1380 cm^−1^ was ascribed to –CH_3_ bending vibration and –CH_2_ twisted vibration. C–C, C–O, and C–H stretching vibrations and bending vibration of C–OH formed obvious absorption peaks in the 1300–800 cm^−1^ region of the spectrum. CTS molecules also displayed a strong and wide absorption peak at 3428 cm^−1^, corresponding to the stretching vibration of O–H and N–H; the absorption peaks at 1660, 1500, and 1425 cm^−1^ corresponded to amide-Ι belt stretching vibration, –NH_2_ deformation vibration, and –CH_2_ and –CH_3_ bending vibrations, respectively. The characteristic absorption peak at 1157 cm^−1^ has been ascribed to the CTS C–O–C asymmetric vibration [33,34].

In a study, the interaction between CTS and starch in starch-based biological foam was shown to cause clear changes in the surroundings of polymer groups, which led to variations in the frequency and intensity of relevant absorption bands, and a unique response was obtained with changes in the characteristic spectra peak wave numbers [35]. Relative to the infrared spectra of pure CTS and starch, the amide peaks of CTS at 1660 cm^−1^ and the characteristic peaks of starch at 1646 cm^−1^ moved to a low-frequency region, and the intensity of the absorption band increased. Correspondingly, the absorption band strength of foam composites at 3413 cm^−1^ increased, which indicates that the hydrogen bond density in the CTS-starch system increased and inter- and intra- molecular hydrogen bonds formed between amino and hydroxyl groups in molecular chains of CTS and starch, thus improving the compatibility between CTS and starch [36,37].

### 3.6. Thermogravimetric Analysis (TGA)

The thermogravimetric curves and corresponding thermogravimetric rate curves of raw materials and starch-based foams are shown in Figure 7. From Figure 7a,b, it was clear that all materials exhibited three stages of heat loss during the whole pyrolysis process. The weight loss in the first stage below 100 °C could be attributed to the evaporation of water. Polysaccharides usually have a strong affinity for water, become hydrated easily, and form large molecules with a messy structure. Hydration properties of these polysaccharides depend on their primary and supramolecular structures [38]. The peak temperatures (Tp, shown in Table 8) for the water evaporation rates of chitosan with different molecular weights were 72.11, 74.70, and 72.17 °C, respectively. Moreover, the foams prepared by chitosan with different molecular weights reached their peaks at these temperatures. The water evaporation rate of chitosan increased with an increase in the molecular weight even though the increment was small. At the same time, the characteristic temperature points of the chitosan–starch foams moved to a higher temperature at this stage. This was due to the cross-linking between chitosan and starch to form new hydrogen bonds that inhibited the increase in water diffusion and thus amplified the water penetration resistance.

The second thermogravimetric stage (between 200 to 400 °C) involved the depolymerization of chitosan molecular chains and the cleavage of starch molecular chains; this included the dehydration of sugar rings and the depolymerization and decomposition of glucose units [39]. Due to the similar depolymerization temperatures of starch and chitosan molecular chains, chitosan–starch foams presented the peak of the mixed weight loss rate of starch and chitosan molecules. The corresponding peak temperatures for the three kinds of foams were 302.88, 305.31, and 305.22 °C. The pyrolysis temperatures of the foams were higher than the temperatures observed for pure starch and chitosan. This indicated that the molecular interaction between the two biomolecules increased the molecular depolymerization temperature. Finally, the high-temperature “tail” between 340 and 500 °C was the third stage of the pyrolysis of chitosan and the vaporization and elimination of starch volatile products [40]. TGA curves showed the high-temperature characteristics of the biological starch-based foams. They depicted that the hydrogen bonding between chitosan and starch resulted in better thermal stability than that observed for neat starch and chitosan. Moreover, the maximum weight loss rate was associated with the molecular weight of chitosan. The value Tp of CTS-4.4/PS was higher than that of the others. This result could be attributed to increased hydrogen bonding and charge interactions.

### 3.7. DSC Analysis

The DSC diagrams of chitosan and starch and CTS/PS foams are shown in Figure 8. The convex peak represents the heat absorption of materials during the glass transition, the intersection point between the peak baseline and the tangent of the rising stage is the initial temperature of glass transition (To), the apex of the peak is the peak temperature (Tp), the peak stage of tangent and baseline by the intersection of terminated for gelation temperature (Tc) and the area of the peak for absorption of heat enthalpy (ΔH); these values are listed in Table 9. As can be seen from Table 9, To, Tp, Tc for chitosan were 34.65, 69.57, 104.72 °C, respectively, and ΔH was 133.53 J/g. The To, Tp, ΔH of porous foams after microwave treatment showed a backward trend with the increase in molecular weight of chitosan. Tester et al. [41] believed that the vitrification transition temperatures (To, Tp, and Tc) were associated with the perfection and stability of crystal. The advancement of glass transition temperature indicated that the temperature required to dissolve the crystal was reduced, indicating the worse stability of the crystal, and the backward shift was the opposite. Therefore, it can be concluded that the molecular weight of chitosan affected the stability of crystals in starch-based foam: the higher the molecular weight of chitosan, the greater the stability of the crystal in foams. The content of crystals could be determined by comparison of enthalpy and the enthalpy of samples increased with the increase in the crystal content. In addition, the improvement of crystal integrity, the enhancement of interaction between amylose or between amylose and amylopectin, the formation of chitosan and starch complex, and the formation of orderly crystalline regions may also increase the enthalpy value [42]. As shown in Table 9, the enthalpy of pure starch is 131.33 J/g, which is higher than that of some chitosan–starch composites; this is due to the easy formation of a double helix structure between short-chain molecules in starch, making the internal structure of starch stronger and more close. CTS/PS foams increased with the increase in the molecular weight of chitosan, which may be caused by the formation of partially ordered crystal regions between starch molecules and chitosan complexes and the longer the molecular chain of chitosan is, the slower its crystal structure is destroyed in the glass transition.

### 3.8. Water Solubility Analysis

Figure 9 shows the solubility of CTS-starch-based foams in deionized water. The overall structure of starch-based foams was uniform and the edges were clearly visible (Figure 8a) when foams were submerged in water. After 3 days, the volume of foams increased and the edges appeared to soften. Furthermore, the turbidity of deionized water increased relative to that in the beginning of the experiment. After 10 days, the exterior foam color turned milky white, but the structure of foams remained intact. The foams began to dissolve in water after 18 days, the volume of foams gradually decreased, and water gradually mixed until foams were completely dissolved after 30 days. The whole dissolution process of CTS-starch-based foams demonstrated that the interaction between CTS and starch molecules can keep the morphological structure of foams intact for 10 days in water, but this interaction is effective for a limited time period.

## 4. Conclusions

Biological foams containing CTS and PS were prepared successfully in a new polytef mold in a microwave oven. The effects of different viscosities and concentrations of CTS and different mass ratios of CTS-starch on the morphological, physicochemical, mechanical, and thermal properties of CTS–PS foams were comprehensively studied. The research conclusions can be summarized as follows:

(1) The viscosity of CTS solution at a particular concentration directly reflects the molecular weight of CTS. The molecular weights of CTS with different viscosities were 3.5 × 10^5^, 4.4 × 10^5^, and 5.2 × 10^5^. The results confirmed that high viscosity corresponds to high relative molecular weight of CTS.

(2) Analysis of morphology, pore size, and density of starch-based biological foams indicated that CTS foams comprising CTS of molecular weight 4.4 × 10^5^ exhibit small pores and low density. Moreover, the foams with a larger proportion of starch demonstrated small pore size and low density. Interestingly, the reverse was found to be true in the case of compressive properties; foams comprising CTS of molecular weight 4.4 × 10^5^ exhibited the highest compressive strength, and foams with a larger proportion of starch exhibited higher compressive strength. In addition, orthogonal experiments were set up to confirm that the compressive performance of foams is highest when the molecular weight of CTS is 4.4 × 10^5^, the mass fraction is 4 wt%, and the mass ratio of CTS-PS is 3/4.2.

(3) The change in the characteristic peaks of the foams composed of CTS and PS in the FTIR diagram indicated that the amino group in CTS interacted with the hydroxyl group in the starch. In TGA and DSC analysis, it could be seen that the interaction between starch and chitosan improved the thermal stability of the foam, and in the water solubility test, the interaction maintained the integrity of the foam’s morphological structure in water within a certain period of time, and the foams completely degraded after 30 days.

## Figures and Tables

**Figure 1 polymers-12-02612-f001:**
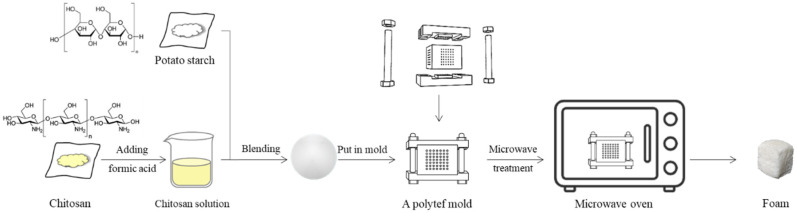
Schematic diagram of preparation of CTS-starch foams.

**Figure 2 polymers-12-02612-f002:**
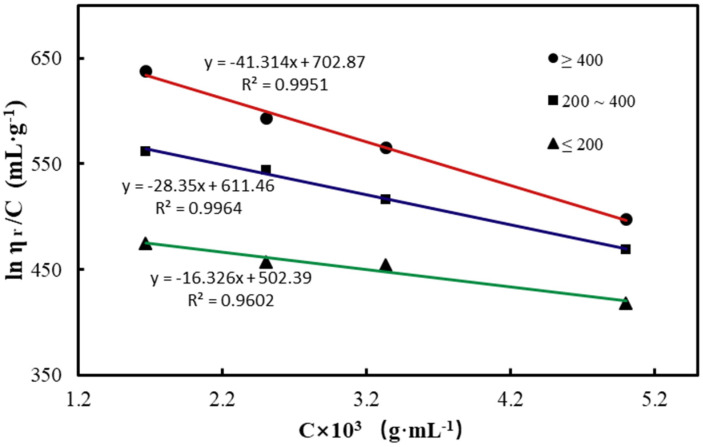
Intrinsic viscosity of CTS in 0.2 M HAc/0.1 M NaAc at 30 °C.

**Figure 3 polymers-12-02612-f003:**
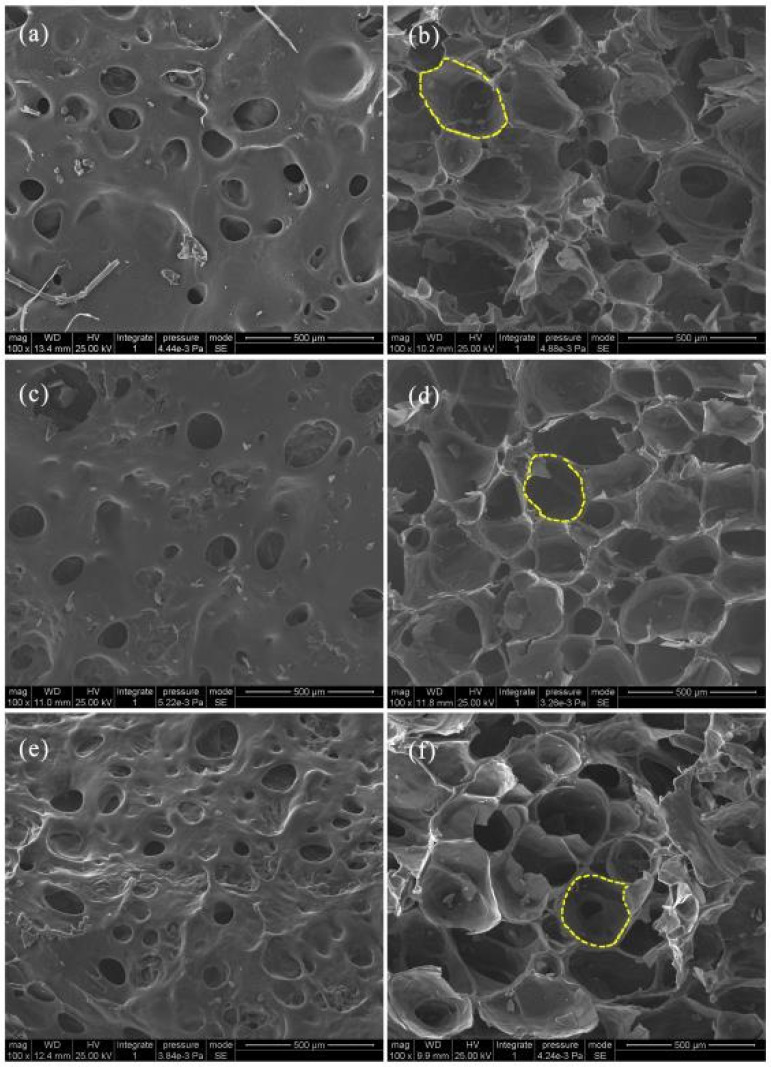
The surface morphologies of foams made from low (**a**), medium (**c**), and high (**e**) viscosity CTS and the corresponding inner morphologies (**b**,**d**,**f**). Scale bar = 500 μm

**Figure 4 polymers-12-02612-f004:**
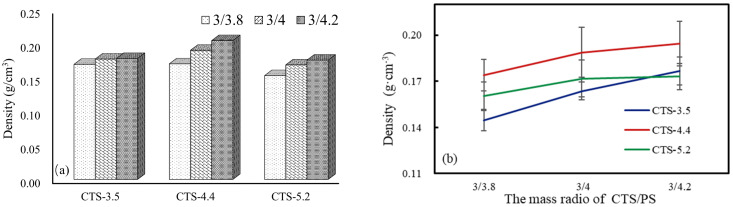
The density of foams with CTS solution for 4.2 wt%.

**Figure 5 polymers-12-02612-f005:**
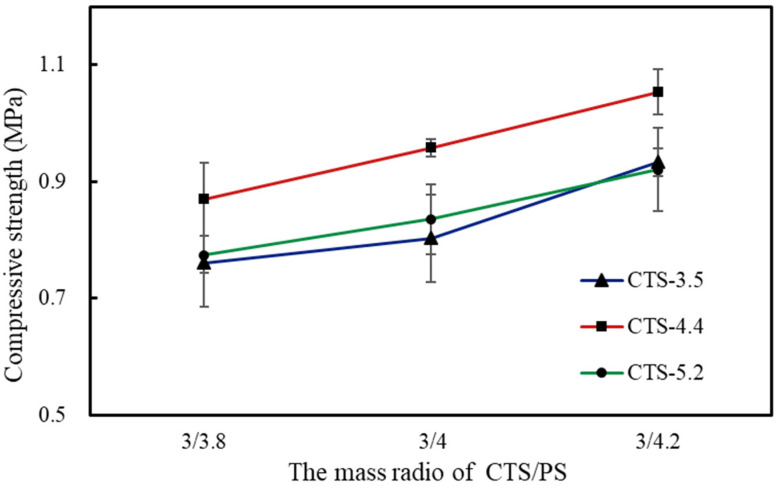
Compressive strength of foams with chitosan solution for 4.2 wt%.

**Figure 6 polymers-12-02612-f006:**
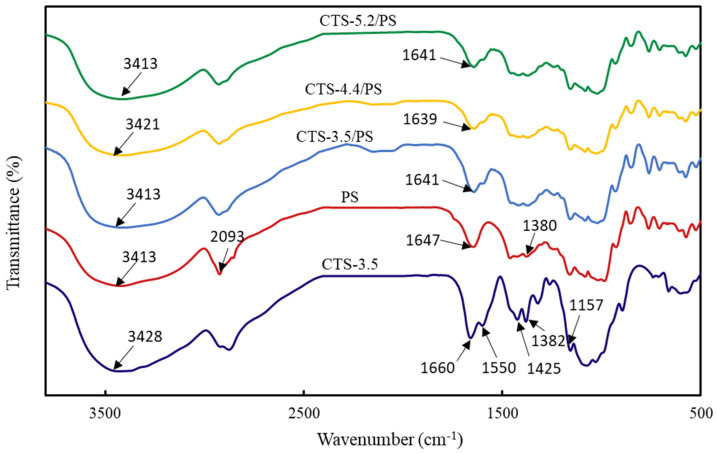
Infrared spectrogram of materials.

**Figure 7 polymers-12-02612-f007:**
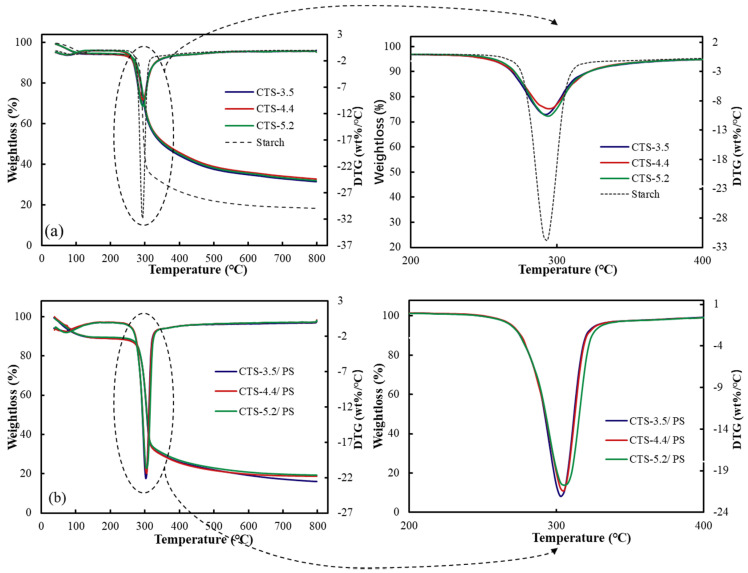
Thermal decomposition process of materials (**a**): raw materials, (**b**): CTS–PS foams.

**Figure 8 polymers-12-02612-f008:**
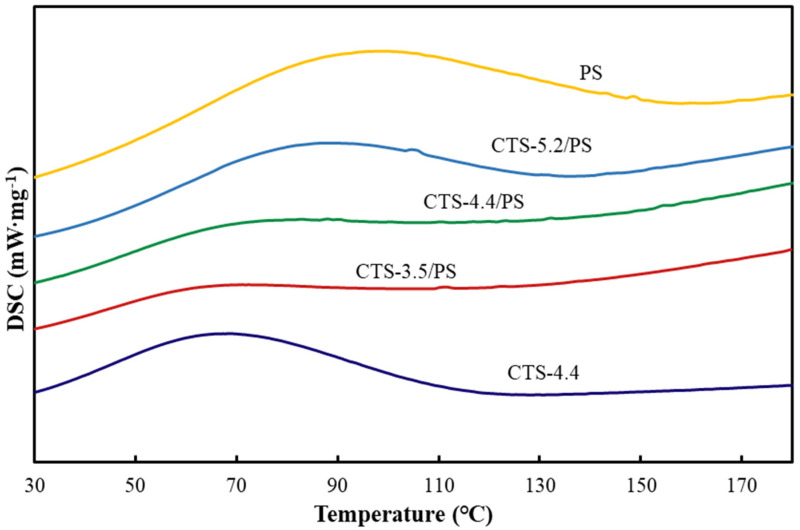
DSC diagrams of raw materials and CTS/PS foams.

**Figure 9 polymers-12-02612-f009:**
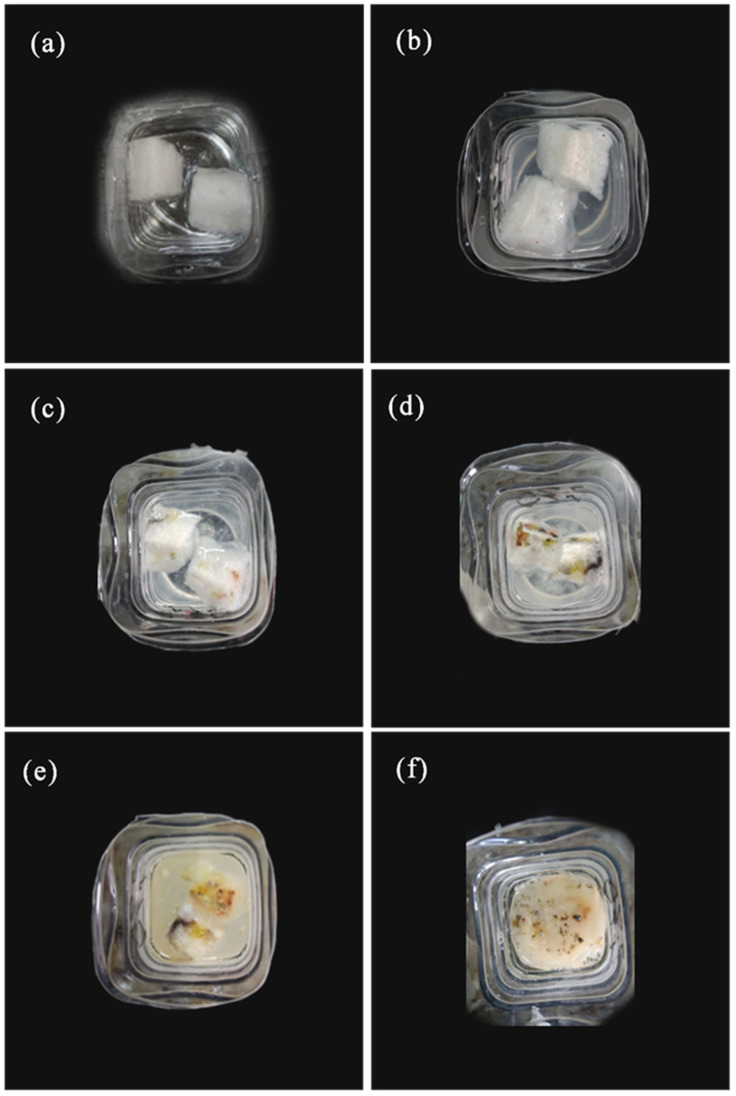
The morphology of CTS-starch-based foams in deionized water (**a**): start time, (**b**): 3 days later, (**c**): 10 days later, (**d**): 18 days later, (**e**): 24 days later, (**f**): 30 days later.

**Table 1 polymers-12-02612-t001:** Formula of CTS-starch foams.

	Factor	Viscosity of CTS (A) (mPa.s)	Mass Fraction of CTS (B) (wt%)	Weight Ratio of CTS/Starch (C) (g)
Level				
1		≤200	3.8	3/3.8
2		200–400	4	3/4
3		≥400	4.2	3/4.2

**Table 2 polymers-12-02612-t002:** Molecular weight of chitosan with different viscosities.

Viscosity (mPa.s)	[*η*] (mL·g^−1^)	R^2^	Molecular Weight (Da)
≤200	502.39	0.995	3.5 × 10^5^
200–400	611.46	0.996	4.4 × 10^5^
≥400	702.87	0.960	5.2 × 10^5^

**Table 3 polymers-12-02612-t003:** The pore size and porosity of foams with CTS solution for 4 wt%.

CTS/PS	CTS-3.5		CTS-4.4		CTS-5.2	
	Pore Size (μm)	Porosity (%)	Pore Size (μm)	Porosity (%)	Pore Size (μm)	Porosity (%)
3/3.8	413.4 (14.0)	69.12	298.8 (24.2)	53.24	407.8 (38.5)	72.29
3/4	392.5 (23.9)	66.98	281.7 (27.0)	55.39	394.2 (54.2)	71.72
3/4.2	391.7 (30.2)	66.12	275.8 (36.3)	51.62	389.8 (28.4)	68.28

**Table 4 polymers-12-02612-t004:** The density values of CTS–PS foams.

Mass Fraction of CTS (wt%)	CTS/PS (g)	CTS-3.5	CTS-4.4	CTS-5.2
	3/3.8	0.170 (0.018)	0.170 (0.008)	0.153 (0.013)
3.8	3/4	0.177 (0.013)	0.190 (0.013)	0.169 (0.008)
	3/4.2	0.178 (0.017)	0.205 (0.037)	0.176 (0.008)
	3/3.8	0.154 (0.018)	0.187 (0.011)	0.151 (0.009)
4	3/4	0.174 (0.010)	0.186 (0.017)	0.169 (0.009)
	3/4.2	0.175 (0.015)	0.197 (0.011)	0.173 (0.008)
	3/3.8	0.144 (0.007)	0.174 (0.010)	0.161 (0.009)
4.2	3/4	0.164 (0.006)	0.189 (0.016)	0.172 (0.012)
	3/4.2	0.177 (0.009)	0.194 (0.015)	0.173 (0.008)

**Table 5 polymers-12-02612-t005:** Compressive strength of starch-based foams.

Samples	CTS-3.8 wt%			CTS-4 wt%			CTS-4.2 wt%		
	3/3.8	3/4	3/4.2	3/3.8	3/4	3/4.2	3/3.8	3/4	3/4.2
CTS-3.5/PS	0.768	0.811	0.983	0.860	0.916	0.958	0.761	0.803	0.934
CTS-4.4/PS	0.847	0.953	1.062	0.938	0.969	1.077	0.870	0.958	1.053
CTS-5.2/PS	0.745	0.792	0.939	0.822	0.892	0.918	0.775	0.836	0.920

**Table 6 polymers-12-02612-t006:** Design and results of orthogonal test.

	Molecular Weight of CTS (A)	Mass Fraction of CTS (B)	Weight Ratio of CTS/PS (C)	Compressive Strength/mPa
1	1	1	1	0.768
2	1	2	2	0.916
3	1	3	3	0.934
4	2	1	2	0.953
5	2	2	3	1.077
6	2	3	1	0.870
7	3	1	3	0.939
8	3	2	1	0.822
9	3	3	2	0.836
K_1_	2.618	2.660	2.460	
K_2_	2.900	2.815	2.705	
K_3_	2.597	2.640	2.950	
k_1_	0.873	0.887	0.820	
k_2_	0.967	0.938	0.902	
k_3_	0.866	0.880	0.983	
R	0.101	0.058	0.163	
Optimal level	A_2_	B_2_	C_3_	
Optimal combination		A_2_B_2_C_3_		

**Table 7 polymers-12-02612-t007:** Variance analysis of orthogonal test.

Factor	Square Sum (SS)	Degree of Freedom (DF)	Mean Square (MS)	F-Value	Significance Probability (P)
A	0.030	2	0.015	40.411	0.024
B	0.005	2	0.002	6.112	0.141
C	0.043	2	0.021	56.613	0.017
Error	0.001	2			

**Table 8 polymers-12-02612-t008:** Thermal decomposition parameters of materials.

Samples	T_1_ ^a^ (°C)	T_P_ (°C)	Residue (%)
CTS-3.5	72.11	292.98	31.44
CTS-4.4	74.70	295.40	32.80
CTS-5.2	72.17	292.94	32.02
PS	92.77	293.64	18.41
CTS-3.5-PS	72.77	302.88	15.97
CTS-4.4-PS	74.22	305.31	18.83
CTS-5.2-PS	73.60	305.22	19.18

^a^ T_1_ = Temperature at the peak below 100 °C on derivative thermogravimetry (DTG) curve, Tp = Temperature at the peak of the whole DTG curve, Residue = Mean value of residual mass.

**Table 9 polymers-12-02612-t009:** The glass transition temperature and enthalpy of samples.

Samples	To (°C)	Tp (°C)	Tc (°C)	ΔH (J/g)
CTS-4.4	34.65	69.57	104.72	133.53
PS	43.67	94.48	153.09	131.33
CTS-3.5/PS	27.93	62.38	104.67	86.32
CTS-4.4/PS	39.02	72.08	126.83	123.91
CTS-5.2/PS	50.96	86.32	116.99	167.86

To = initial temperature of glass transition, Tp = the apex of the peak is the peak temperature, Tc = the peak stage of tangent and baseline by the intersection of terminated for glass transition temperature; ΔH = the area of the peak for absorption of heat enthalpy.

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
