# Peer review of "Biodegradable Starch/Chitosan Foam via Microwave Assisted Preparation: Morphology and Performance Properties"

_polymers, 2020, doi:10.3390/polym12112612_

Round 1
Reviewer 1 Report
Recommendation: major revision
Comments to Authors: Xian Zhang, Zhuangzhuang, Teng and Runzhou Huang
Manuscript Number: polymers-988245
Article Type: Article
Article Title: Biodegradable starch/chitosan foam via microwave treatment: Morphology and Performance Properties
Overview and general recommendation
The introduction provides a sufficient background and contains relevant references to the problem raised.
The methods were presented correctly and chronologically adequate to the conducted research and are adequately described.
The results need to be improved.
Please let me know about Authors, looks like error with Zhuangzhuang
Major comments:
No statistics of the results were developed in the manuscript; please in the chapter methods put the information about statistical method and calculate for the results in tables. After that please discuss the results again.
Overall Recommendation
My recommendation is major revision
Author Response
Dear editor and reviewers:
Thank you very much for your time and efforts in reviewing our manuscript. We sincerely appreciate your perspicacious suggestions, which have helped us greatly in improving the research and manuscript. We provide this response letter to explain the details of our revisions in the manuscript and our responses to the reviewers’ comments as follows. In order to make the changes easily viewable for you and the reviewers, in the revised paper, we marked the revision with red color. Besides, we have carefully checked through the whole manuscript and corrected some words and grammar mistakes. We hope the revised paper would satisfy you and the reviewers.
Overview and general recommendation
The introduction provides a sufficient background and contains relevant references to the problem raised.
The methods were presented correctly and chronologically adequate to the conducted research and are adequately described.
The results need to be improved.
Major comment: No statistics of the results were developed in the manuscript; please in the chapter methods put the information about statistical method and calculate for the results in tables. After that please discuss the results again.
Respond: We all members extremely appreciate you for your valuable comments. In the introduction, there is an urgent demand for a new biodegradable plastic to replace the plastic made from petroleum in the world, which leading to a lot of efforts made by many researchers in this field. Then we put forward a kind of biodegradable foam material prepared by microwave radiation from the previous works.
In terms of the research design and methods, we adopted a comprehensive experiment of 27 groups in total with three factors and three levels (without considering the experimental repeatability), and judged performances of various aspects of the starch-based foams through the data results obtained from tests(Table.6).
On the basis of comprehensive experiment, the orthogonal test analysis and multi-factor analysis of variance were carried out to explore the interaction factors influence on mechanical properties (the results and discussion were shown in Line 309), which will further help the experiment to determine the best process parameter of each level, and to determine the relative importance of individual parameters and the combination of process parameters with high performance(Table.7). In the section of experimental design description, we made further modifications to help readers read more clearly.
Please let me know about Authors, looks like error with Zhuangzhuang
Respond: Many thanks for this comment. We apologized for our mistake . The second author is Zhuangzhuang Teng ,the first name is Zhuangzhuang ,the last name is Teng. We revised the name spelling on the author list.
Yours sincerely,
Corresponding author: Runzhou Huang, runzhouhuang@njfu.edu.edu
Reviewer 2 Report
what does " Biodegradable foam via microwave treatment" mean to readers? synthesis, applications??? please revise.
Fig. 1 should be revised. it is not good.
3.6 TG Analysis should be TGA. please correct in the manuscript.
Fig. 7 The curves are not well distinguishable. Please use dash line for some of them
The 1st paragraph of the introduction should be removed as it is well known. The manuscript lacks application. At least is should be well proposed (no need of new tests). such foam may be used for tissue regeneration or water treatment. Please add in the introduction. Recommended litratures (e.g., ACS Appl. Nano Mater. 2020, 3, 7, 6210–6238; Carbo Poly, 212, 2019, 450)
References: There are some old references. Please remove/replace the outdated references. Some of them:
8. Wan, Y.; Creber, K.A.M.; Peppley, B.; Bui, V.T. Ionic conductivity of chitosan membranes. Polymer 2003, 44, 470 1057-1065.
471 19. Tonhi, E.; Plepis, A.M.D. Preparation and characterization of collagen-chitosan blends. Quim Nova 2002, 25, 472 943-948
39. Fang, J.M.; Fowler, P.A.; Tomkinson, J.; Hill, C.A.S. The preparation and characterisation of a series of 512 chemically modified potato starches. Carbohydrate Polymers 2002, 47, 245-252.
41. R. F. Tester, W. R. Morrison, Swelling and gelatinization of cereal starches. II, Waxy rice starches. Cereal 516 Chemistry.1990, 67, 558-563.
Author Response
Dear editor and reviewers:
Thank you very much for your time and efforts in reviewing our manuscript. We sincerely appreciate your perspicacious suggestions, which have helped us greatly in improving the research and manuscript. We provide this response letter to explain the details of our revisions in the manuscript and our responses to the reviewers’ comments as follows. In order to make the changes easily viewable for you and the reviewers, in the revised paper, we marked the revision with red color. Besides, we have carefully checked through the whole manuscript and corrected some words and grammar mistakes. We hope the revised paper would satisfy you and the reviewers.
Responds to Reviewer #2:
Comment 1: What does " Biodegradable foam via microwave treatment" mean to readers? synthesis, applications??? please revise.
Respond: Many thanks for this comment. Microwave treatment is a new heating technology to prepare the starch-based foams. Conventional heating is a way to heat the surrounding or surface of solid through conduction, convection and radiation heat transfer, so that the surface of solid gets heat, and then conducts the heat inside solid internal by means of heat conduction. Hot air, Furnace gas, superheated steam, or far infrared radiation, etc. can be used as a heating medium and this heating method has low efficiency and long heating time. Microwave heating is a new heating method and when it produces and comes into contact with objects, it is not hot air, but electromagnetic energy. As long as there is microwave radiation, the material can be heated immediately, which is a very fast heating way. The electromagnetic wave has strong penetrating power and can be heated uniformly inside and outside. Microwave heating has less heat loss and higher utilization of microwave energy. This is why we chose microwave heating and we used microwave heating to make a mixture of starch and chitosan to form a foam that could be degraded in this manuscript.
According to your comment, the title has been revised to “Biodegradable starch/chitosan foam via microwave assisted preparation: Morphology and Performance Properties”.
Comment 2: Fig. 1 should be revised. it is not good.
Respond: Thank you for this advice. We revised this figure in the manuscript.
Comment 3: 3.6 TG Analysis should be TGA. please correct in the manuscript.
Respond: Many thanks for this remind. We corrected this in the revised manuscript.
Comment 4: Fig. 7 The curves are not well distinguishable. Please use dash line for some of them.
Respond: Thank you for your remind. We added some dotted lines to distinguish them in the manuscript.
Comment 5: The 1st paragraph of the introduction should be removed as it is well known. The manuscript lacks application. At least is should be well proposed (no need of new tests). such foam may be used for tissue regeneration or water treatment. Please add in the introduction. Recommended literatures (e.g., ACS Appl. Nano Mater. 2020, 3, 7, 6210–6238; Carbo Poly, 212, 2019, 450)
Respond: Thank you for this advice. White pollution is a very serious problem in China, especially the oil resources are decreasing day by day. How to reduce white pollution and find a kind of biodegradable plastic is very feasible, which fits well with the current theme of environmental protection, and highlights the importance of this research. Biodegradable and environmentally friendly materials will have great development potential in the future, and our contribution to environmental protection from this research is particularly useful.
In addition, we discussed the feasibility of chitosan/starch-based foam as a biomedical and food packaging applications in the introduction, and quoted the very useful literatures you recommended, which makes this manuscript more informative.
Comment 6: References: There are some old references. Please remove/replace the outdated references. Some of them:
- Wan, Y.; Creber, K.A.M.; Peppley, B.; Bui, V.T. Ionic conductivity of chitosan membranes. Polymer 2003, 44, 470 1057-1065.
- Tonhi, E.; Plepis, A.M.D. Preparation and characterization of collagen-chitosan blends. Quim Nova 2002, 25, 472 943-948
- Fang, J.M.; Fowler, P.A.; Tomkinson, J.; Hill, C.A.S. The preparation and characterisation of a series of 512 chemically modified potato starches. Carbohydrate Polymers 2002, 47, 245-252.
- R. F. Tester, W. R. Morrison, Swelling and gelatinization of cereal starches. II, Waxy rice starches. Cereal 516 Chemistry.1990, 67, 558-563.
Respond: Many thanks for this advice. We have deleted the references 8 and 19, and selected a new literature to replace the reference 39, but unfortunately, the reference 41 cannot be deleted or replaced, it is essential in this manuscript.
Yours sincerely,
Corresponding author: Runzhou Huang, runzhouhuang@njfu.edu.edu
Round 2
Reviewer 1 Report
I accept